# Combining APHRODITE Rain Gauges-Based Precipitation with Downscaled-TRMM Data to Translate High-Resolution Precipitation Estimates in the Indus Basin

Rabeea Noor [1,†], Arfan Arshad [2,3,*,†], Muhammad Shafeeque [4], Jinping Liu [5,6,7], Azhar Baig [1,8], Shoaib Ali [9], Aarish Maqsood [1], Quoc Bao Pham [10], Adil Dilawar [11,12], Shahbaz Nasir Khan [13], Duong Tran Anh [14,15] and Ahmed Elbeltagi [16]

1 Department of Agricultural Engineering, Bahauddin Zakariya University, Multan 60800, Pakistan
2 Department of Biosystems and Agricultural Engineering, Oklahoma State University, Stillwater, OK 74078, USA
3 Department of Irrigation and Drainage, Faculty of Agricultural Engineering and Technology, University of Agriculture Faisalabad, Faisalabad 38000, Pakistan
4 Climate Lab, Institute of Geography, University of Bremen, 28359 Bremen, Germany
5 College of Surveying and Geo-Informatics, North China University of Water Resources and Electric Power, Zhengzhou 450046, China
6 Key Laboratory of Hydrology-Water Resources and Hydraulic Engineering, Hohai University, Nanjing 210098, China
7 Hydraulics and Geotechnics Section, KU Leuven, Kasteelpark Arenberg 40, BE-3001 Leuven, Belgium
8 Department of Bioresource Engineering, McGill University, 21111 Lakeshore, Ste Anne de Bellevue, QC H9X 3V9, Canada
9 School of Water Conservancy & Civil Engineering, Northeast Agricultural University, Harbin 150030, China
10 Faculty of Natural Sciences, Institute of Earth Sciences, University of Silesia in Katowice, Będzińska Street 60, 41-200 Sosnowiec, Poland
11 State Key Laboratory of Resources and Environment Information System, Institute of Geographic Sciences and Natural Resources Research, Chinese Academy of Sciences, Beijing 100101, China
12 University of Chinese Academy of Sciences (UCAS), Beijing 100049, China
13 Department of Structures and Environmental Engineering, University of Agriculture Faisalabad, Faisalabad 38000, Pakistan
14 Laboratory of Environmental Sciences and Climate Change, Institute for Computational Science and Artificial Intelligence, Van Lang University, Ho Chi Minh City 700000, Vietnam; duong.trananh@vlu.edu.vn
15 Faculty of Environment, Van Lang University, Ho Chi Minh City 700000, Vietnam
16 Agricultural Engineering Department, Faculty of Agriculture, Mansoura University, Mansoura 35516, Egypt
* Correspondence: aarshad@okstate.edu
† These authors contributed equally to this work and should be considered co-first authors of this paper.

**Abstract:** Understanding the pixel-scale hydrology and the spatiotemporal distribution of regional precipitation requires high precision and high-resolution precipitation data. Satellite-based precipitation products have coarse spatial resolutions (~10 km–75 km), rendering them incapable of translating high-resolution precipitation variability induced by dynamic interactions between climatic forcing, ground cover, and altitude variations. This study investigates the performance of a downscaled-calibration procedure to generate fine-scale (1 km × 1 km) gridded precipitation estimates from the coarser resolution of TRMM data (~25 km) in the Indus Basin. The mixed geographically weighted regression (MGWR) and random forest (RF) models were utilized to spatially downscale the TRMM precipitation data using high-resolution (1 km × 1 km) explanatory variables. Downscaled precipitation estimates were combined with APHRODITE rain gauge-based data using the calibration procedure (geographical ratio analysis (GRA)). Results indicated that the MGWR model performed better on fit and accuracy than the RF model to predict the precipitation. Annual TRMM estimates after downscaling and calibration not only translate the spatial heterogeneity of precipitation but also improved the agreement with rain gauge observations with a reduction in RMSE and bias of ~88 mm/year and 27%, respectively. Significant improvement was also observed in monthly (and daily) precipitation estimates with a higher reduction in RMSE and bias of ~30 mm mm/month (0.92 mm/day) and 10.57% (3.93%), respectively, after downscaling and calibration procedures. In

general, the higher reduction in bias values after downscaling and calibration procedures was noted across the downstream low elevation zones (e.g., zone 1 correspond to elevation changes from 0 to 500 m). The low performance of precipitation products across the elevation zone 3 (>1000 m) might be associated with the fact that satellite observations at high-altitude regions with glacier coverage are most likely subjected to higher uncertainties. The high-resolution grided precipitation data generated by the MGWR-based proposed framework can facilitate the characterization of distributed hydrology in the Indus Basin. The method may have strong adoptability in the other catchments of the world, with varying climates and topography conditions.

**Keywords:** RF; MGWR; TRMM; spatial downscaling; calibration; rain gauges

## 1. Introduction

Precipitation is an important part of the hydrological cycle which plays a critical role in regulating land surface and hydrologic processes, including stream flow, groundwater recharge, soil moisture, and atmospheric temperature in the earth's system [1–4]. Precipitation variability either in the form of wetness or droughts [5], could adversely affect the water resources, agriculture, and other ecosystem services [6–8]. High-resolution and precise information on precipitation is extremely important for a wide range of applications, particularly to address spatial–temporal distribution of hydrology [9–12]. Precipitation observations from rain-gauge stations generally have good accuracy [10]; however, sparse distribution of rain gauges especially in the large catchments such as the Indus Basin hinders their regional-scale applications to conduct research, policies, and decision-making processes for water resources management. On-going space observations from the past few decades made it possible to monitor precipitation with global and regional coverage [13,14] such as the Tropical Rainfall Measuring Mission (TRMM) [15–17], Global Satellite Mapping of Precipitation (GSMaP) [18], Global Precipitation Climatology Project (GPCP) [19], Climate Prediction Center (CPC) [20], MORPHing technique (CMORPH) [21], Precipitation Estimation from Remotely Sensed Information using Artificial Neural Networks (PERSIANN), PERSIANN Cloud Classification System (PERSIANN-CCS) [22], Integrated Multi-satellite Retrievals for GPM (IMERG) available at different spatial–temporal resolutions [23]. Among these, the TRMM-based precipitation estimates have commonly been utilized for basins to catchments scale to monitor land-hydrological processes [24]. Despite these efforts, TRMM data are still criticized due to their coarse resolution [10,25], rendering them incapable of translating large-scale variability in precipitation induced by dynamic interactions between climatic forcing, ground cover, and altitude variations. Thus, to achieve the fine-scale hydrological assessment of the precipitation cycle, improving the spatial resolution of TRMM precipitation data is highly demanded in the study region [10,26–28].

The spatial downscaling methods enhance the spatial resolution of grided satellite precipitation products [10,28–31], making it more applicable for sub-basins studies, and local-scale applications [10,32,33] to study ecology and hydrological cycles, discharge simulations, and analyze soil erosion [34]. Statistical and dynamic downscaling are two main approaches to improving the spatial information in gridded products [10,28,35]. Although the dynamic downscaling method can be useful in understanding complex physical processes (i.e., atmosphere, ocean, and land surface), it is limited in applicability because it is computationally intensive and time-consuming. Moreover, error chances are relatively higher [36,37].

The statistical downscaling approach uses simplified algorithms to estimate fine-scale precipitation at a given location based on high-resolution explanatory variables (e.g., temperature, evapotranspiration, vegetation, soil moisture, etc.) [38] and is a commonly applied approach around different watersheds of the world [10,26,32,38,39]. The most often utilized statistical downscaling approaches contain linear and nonlinear models such as GWR (Geographical Weighted Regression), PLSR (partial least squares regression), RF

(Random Forest) model, and ANN (Artificial Neural Networks) [10,22,40–42]. Studies have demonstrated the superiority of the localized GWR model in comparison to the linear and nonlinear regressions sole regressions [38,43] and machine learning techniques (e.g., ANN and RF) [44]. GWR applications are still criticized because it assumes the non-stationarity of regression coefficients for all explanatory variables [45,46], while the previous studies [45,47,48] have reported that regression coefficients could be stationary ("that do not change in space") for some variables and non-stationary ("varied over space") for other explanatory variables. Introducing all explanatory variables in the GWR model with non-stationary regression coefficients could reduce the model fitting and result in uncertainty in the final prediction [45,49,50]. To overcome this constraint, the MGWR (mixed geographically weighted regression) model handles explanatory variables with stationary and non-stationary coefficients [45]. This approach isolates the spatially varying (non-stationary) and constant (fixed terms/stationary) regression coefficients of explanatory variables to enhance the model fitting and accuracy [45]. Therefore, this study uses the MGWR model to downscale the coarse-resolution TRMM data to high resolution. However, applying a localized linear MGWR model to downscale the TRMM data is still lacking for the entire Indus Basin. Further benchmarking of the research comparing MGWR with a nonlinear reference model (e.g., RF model) is therefore warranted, which our study has reported. Random Forests (RF) [51] is a popular machine learning approach that is known for being able to handle nonlinear relationships between explanatory variables without overfitting and has shown excellent applications in watershed studies [52,53]. The current downscaling approach used the original TRMM data as input along with high-resolution explanatory variables to predict high-resolution precipitation estimates. Therefore, errors associated with model input parameters, including TRMM, could induce uncertainties in the final downscaled results [54]. Therefore, adequate data-merging techniques are required to improve the reliability of model-based estimated precipitation via incorporating downscaled precipitation with ground observation records [45,50,55,56]. The geographical difference analysis (GDA) and geographical ratio analysis (GRA) are the most utilized approaches for merging ground observation records with satellite data [57,58]. However, according to the literature, the GRA approach performed better than GDA [45,46].

This study presents a methodology to translate the high-resolution spatial–temporal precipitation estimates by combining downscaled TRMM data with APHRODITE's (Asian Precipitation—Highly-Resolved Observational Data Integration Towards Evaluation) rain gauge-based data over the Indus Basin. The central hypothesis of this study is that MGWR-based downscaling and calibration procedures can translate spatial heterogeneity in precipitation variability and improve the precision of precipitation estimated. The main objectives of this study are twofold: (1) benchmarking the performance of MGWR in comparison to the nonlinear reference model (e.g., RF model) to predict the high-resolution precipitation forecasts, and (2) Demonstrating how a smart combination of downscaled TRMM data with APHRODITE rain gauge-based precipitation can improve precision in precipitation estimates.

## 2. Materials and Methods

### 2.1. Study Area

The study area covers the Indus Basin, which shares drainage areas in four countries (China, India, Afghanistan, and Pakistan), with Pakistan having the largest share of the water. It is the 12th largest watershed worldwide, occupying an area of 1.12 million km$^2$ [59]. The study region contains the high Hindu Kush, Karakoram, Himalaya, and Tibetan Plateau Mountain ranges. The Indus River is the largest in the Indus Basin, which originates from the Tibetan Plateau in China and flows into the Arabian Sea. Glaciers cover about 220,000 km$^2$ in the study area. The elevation of the Indus Basin ranges between 0 and >8000 m above sea level (Figure 1a). There are 60 rain gauge stations with available observations scattered across the Indus Basin. Rain gauges are divided into three different groups based on the elevation changes e.g., elevation zone 1 (0–500 m), zone 2 (5000–1000 m), and

zone 3 (>1000 m). The number of gauges along zone 1, zone 2, and zone 3 are 30, 7, and 23, respectively. Precipitation is subjected to great spatial variability throughout the region, with ranges as low as 150 mm in the south to 1200 mm in the highland areas (Figure 1b). Precipitation gradually increases up to 1000 m and decreases with a nonlinear distribution across the high-altitude regions (Figure 1c). The highest annual precipitation is recorded as 1673 mm/year over the Muree stations, followed by Rawalkot (1568 mm/year).

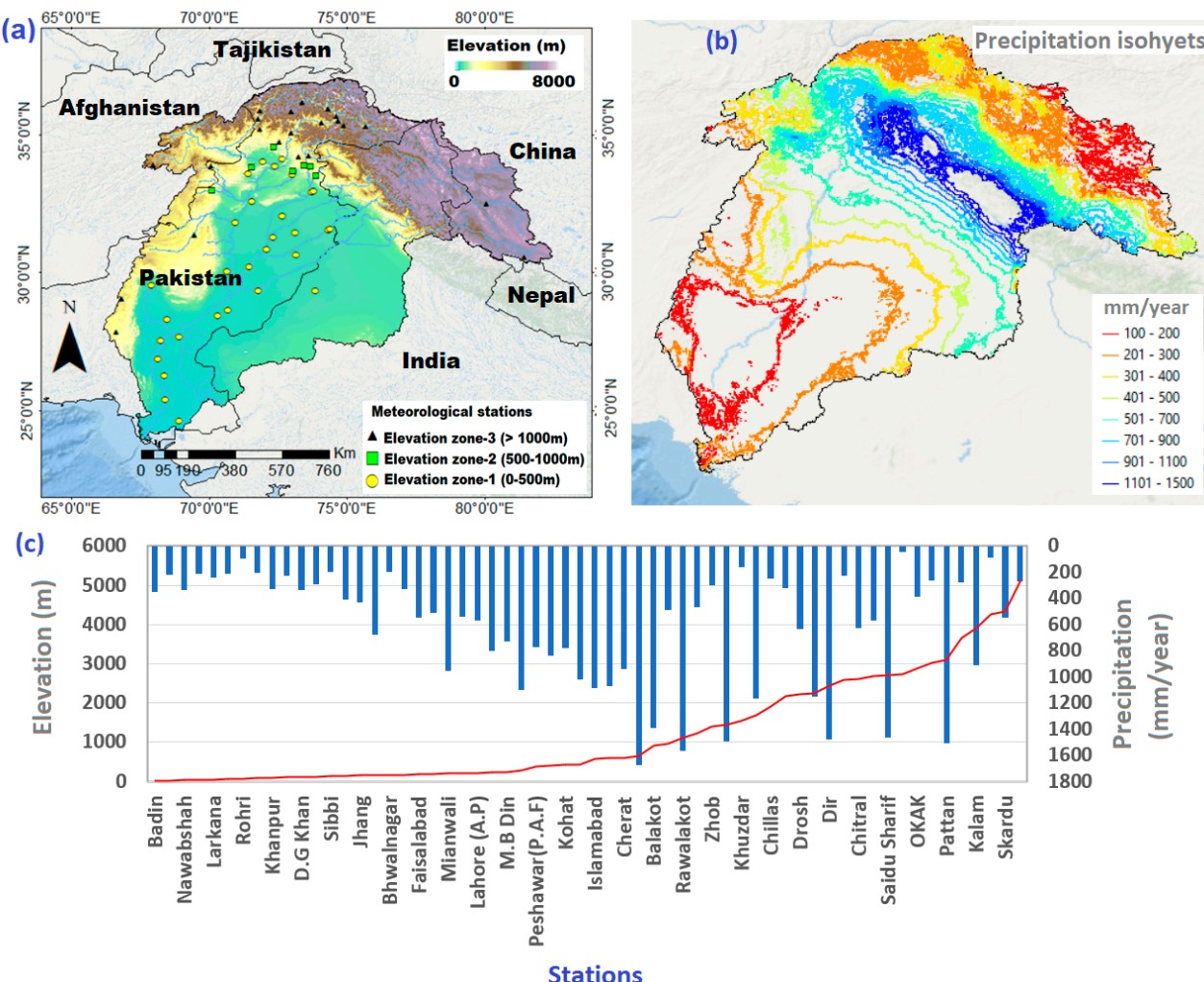

**Figure 1.** (**a**) Elevation map of Indus Basin, rivers network, and meteorological stations corresponding to three different elevation zones, (**b**) annual precipitation isohyets, and (**c**) annual precipitation across 60 rain gauge stations along the elevation profile of 0–6000 m.

### 2.2. Data Sources and Variables

To support the methods and objectives of this study, we have collected data from multiple sources including satellite, reanalysis, and rain gauge observations (Table 1). More details are described as follows.

**Table 1.** A detailed description of datasets utilized in the study.

| Variables | Versions | Resolution | Period | Sources |
|---|---|---|---|---|
| Precipitation | TRMM_3B42 | ~25 km (daily) | 2000–2019 | https://disc.gsfc.nasa.gov/mirador-guide (accessed on 23 November 2021) |
| ET | MOD16 | 1 km × 1 km (monthly) | 2000–2019 | https://www.ntsg.umt.edu/project/modis/mod16.php#data-product (accessed on 23 November 2021) |
| NAVI | MOD13A3 | 1 km × 1 km (monthly) | 2000–2019 | https://lpdaac.usgs.gov/dataset_discovery/modis (accessed on 23 November 2021) |
| LST | MOD11A2 | 1 km × 1 km (8 days intervals) | 2000–2019 | NASA Land Processes Distributed Active Archive Center |
| Elevation | SRTM | 90 m × 90 m | | http://www2.jpl.nasa.gov/srtm/ (accessed on 23 November 2021) |
| Cloud cover | ERA-5 | 0.125° × 0.125° (monthly) | 2000–2019 | http://apps.ecmwf.int/datasets/data/interim-full-moda/ (accessed on 26 November 2021) |
| Wind speed | ERA-5 | 0.125° × 0.125° (monthly) | 2000–2019 | http://apps.ecmwf.int/datasets/data/interim-full-moda/ (accessed on 29 November 2021) |
| Weather stations data | Ground-observations | stations (daily) | 2007–2012 | PMD (Pakistan Metrological department) |
| Aphrodite | APHRO_MA_V110 | 0.25° × 0.25° (daily) | 2007–2012 | climatedataguide.ucar.edu/data-formats/netcdf |

TRMM precipitation dataset: The TRMM (Tropical Rainfall Measuring Mission) is a joint mission launched in November 1997 by NASA (National Aeronautics and Space Administration) and JAXA (Japan Aerospace Exploration Agency), which offers precipitation records with global coverage. We used daily precipitation data of TMPA (TRMM Multi-satellite Precipitation Analysis) 3B42 version 7 which is available at 0.25° × 0.25° spatial resolution. This daily accumulated precipitation product is generated from the research-quality 3-hourly TRMM Multi-Satellite Precipitation Analysis TMPA (3B42). The product is already processed by NASA Goddard Earth Science Data (https://disc.gsfc.nasa.gov/datasets/TRMM_3B42_Daily_7/summary?keywords=TMPA, accessed on 23 November 2021).

APHRODITE rain gauge-based precipitation: The daily gridded (0.25° × 0.25°) precipitation data of the densely rain gauge-based dataset (APHRIDTE) are taken from version (APHRO_MA_V110). They are available for sub-domains Monsoon Asia, Middle East, Russia, and Japan. It has been widely used as a calibration baseline and in other hydro-climate studies in Asian watersheds [60–63]. APHRODITE precipitation data were used to enhance the precision of TRMM data.

Observational rain gauge precipitation records: Daily weather data for 60 rain gauge stations from 2007–2012 were obtained from the PMD (Punjab Meteorological Department), which was only used for validation of downscaled results for precipitation estimates.

Explanatory variables: The high-resolution (1 km × 1 km) explanatory variables (i.e., Evapotranspiration (ET), wind speed, elevation, slope, normalized difference vegetation index (NDVI), cloud cover, land surface temperature (LST)) are used in the downscaling procedure. Except for the elevation and slope, all datasets were collected from the period 2000 to 2019. The detail of the datasets is explained in Table 2. Elevation data were acquired from SRTM (Shuttle Radar Topography Mission) at 90 m × 90 m, which is resized to 1 km × 1 km resolution utilizing the cubic interpolation technique. Further, the slope map was prepared from the elevation data using the slope tool in ArcGIS. The monthly NDVI was acquired from the Terra MODIS sensor onboard NASA's satellite, provided by the LPDAC (Land Processes Distributed Active Archive Center) at 1 km × 1 km resolution. The LST data at 1 km × 1 km resolution is also acquired from the LPDAC, provided by the MODIS onboard sensor (version MOD11A2) at eight days and the nighttime interval, which is converted into the monthly mean by taking an average of 8 days records. The actual ET of MODIS was acquired from the Numerical Terra dynamic Simulation Group at 1 km × 1 km resolution. Monthly cloud cover and wind speed data were acquired from the European Center for Medium-Range Weather Forecasts (ECMWF), which were resampled at 0.01° × 0.01° resolution using the cubic interpolation technique in ArcGIS. More details about data collection and preprocessing of seven explanatory variables are also available in an earlier study published by [45].

**Table 2.** List of evaluation matrices and their formulae used in this study. Here, n indicates the number of data samples utilized in the study; $ET_{ip}$ denotes the precipitation estimates of satellites (TRMM, downscaled, and calibrated-downscaled); $ET_{iQ}$ depicts the observed precipitation data from rain gauge stations.

| Statistical Indicators | Formulas |
|---|---|
| Coefficient of determination ($R^2$) | $\left[ \dfrac{\sum_{i=1}^{n}\left(ET_{ip}-\overline{ET_p}\right)\left(ET_{iQ}-\overline{ET_Q}\right)}{\sqrt{\sum_{i=1}^{n}\left(ET_{ip}-\overline{ET_p}\right)^2\ \sum_{i=1}^{n}\left(ET_{iQ}-\overline{ET_Q}\right)^2}} \right]^2$ |
| RMSE | $\sqrt{\dfrac{1}{n}\ \sum_{i=1}^{n}\left(ET_{ip}-ET_{iQ}\right)^2}$ |
| Mean square error (MSE) | $\dfrac{1}{n}\sum_{i=1}^{n}\left(ET_{ip}-ET_{iQ}\right)^2$ |
| Bias | $\dfrac{\sum_{i=1}^{n}\left[ET_{ip}(x_i)-ET_{iQ}(x_i)\right]}{\sum_{i=1}^{n}\left[ET_{iQ}(x_i)\right]}\times 100$ |

*2.3. Description of Numerical Models*

We used two computational models, MGWR and RF, to downscale the TRMM data from coarse resolution to fine-scale using explanatory variables as input.

Random Forest (RF) model: RF is an ensemble machine learning model which constructs an optimal model through a regression tree and then makes its final prediction based on the results from all the models [64] (See Figure S1 in the Supplementary Material). The model splits the original data into subdivisions and develops sub-regression models on each subdivision representing a single tree. Outputs from all sub-models are averaged to make a final prediction. The RF model used here could be described as follows.

$$\text{TRMM}_{\text{precipitation}} = f(x) + \varepsilon \text{ where } x = (x_1, x_2, x_3, \ldots, x_n) \tag{1}$$

where $\text{TRMM}_{\text{precipitation}}$ = TRMM based precipitation estimates, $\varepsilon$ = model error estimates, $x = x_1, x_2, x_3, \ldots, x_n$ represent the input variables (i.e., elevation, slope, NDVI, ET, etc., as background fields), and n is the total number of inputs (e.g., seven in our case). Random Grid Search (RGS) is utilized through many iterations to determine the optimal values of the model's parameters. The optimal values of nodes, ntree (number of tree), mtry (number of variables randomly sampled and nodes) were bootstrapped with 2, 250, and 4, respectively.

Mixed Geographically Weighted Regression (MGWR) model: the MGWR model is an extension of GWR, which hold mixed coefficients, e.g., stationary (constant over space) and non-stationary (spatial varying) for explanatory variables and is mathematically described as follows [50,65]:

$$Y_i = \left[ \sum_{j=1}^{k} \alpha_j \, x_{ij} \right] + \left[ \sum_{l=k+1}^{P} \beta_l(u_i, \, v_i) x_{il+\varepsilon(u_i, \, v_i)} \right] \tag{2}$$

where $Y_i$ represents the *i*th observation of TRMM-based precipitation estimates, $\alpha_j$ represents the stationary coefficient of *j*th explanatory variable where j = 1, 2, $\ldots$, k, $x_{ij}$ is *i*th observation of *j*th stationary explanatory variable, $\beta_l(u_i, \, v_i)$ is the non-stationary coefficients of *l*th explanatory variable at location $(u_i, \, v_i)$ where $l = 1, 2, \ldots, p$, $\varepsilon(u_i, \, v_i)$ is the models error estimates at each location $(u_i, \, v_i)$, and $x_{il}$ is the ith observation of the *l*th local explanatory variable. The geographical variability test (GVT) is performed to explore the stationary and non-stationary coefficients for explanatory variables. Firstly, the traditional GWR model is run keeping all explanatory variables as non-stationary. Secondly, different sets of new sub-models were built by keeping one variable stationary while remaining non-stationary. If the new sub-models performed with higher fitting than the traditional GWR model, the AICc (Akaike information criterion) values indicate that the model comparison criterion would be lower, while DIFF (i.e., Difference of Criterion) would be positive, indicating that switched variables in sub-models are stationary and contrariwise [45,49,65]. In general, if the sub-models have DIFF criterion values > 2, then the coefficients of variables do not vary spatially and should be introduced as stationary. In contrast, if the DIFF < −2, the coefficients of variables should be introduced as spatially varying (non-stationary) terms in the model [65]. The procedure is repeated until we confirm the stationary and non-stationary coefficients of all explanatory variables and build an MGWR model with different settings of predictor variables following Equation (2) (see more details in Figure S2 in Supplementary Material).

The regression coefficients (intercept and slope of each variable) in Equation (2) were estimated on each location using the following matrix:

$$\beta(u_i, \, v_i) = (x^T(w(u_i, \, v_i))x)^{-1}x^Tw((u_i, \, v_i))y \tag{3}$$

where:

$\beta(u_i, \, v_i)$ = local regression coefficients to be estimated at the location $(u_i, \, v_i)$

x = explanatory variables

y = dependent variable

w (u$_i$, v$_i$) = weight matrix to be estimated at the location (u$_i$, v$_i$)

Solving the weight matrix required the proper selection of kernel filters which play an important role in model fitting. We applied a bi-square (BI) filter based on their higher accuracy, as suggested in the previous studies [45,50]. The continuous weights (w$_{ij}$) were estimated with a bi-square (BI) filter using the following expression [66]

$$w_{ij} = \begin{cases} 1 - \left( \frac{d_{ij}^2}{\theta_{i(k)}} \right)^2, & d_{ij} < \theta_{i(k)} \\ 0, & d_{ij} > \theta_{i(k)} \end{cases} \tag{4}$$

where:

$w_{ij}$ = Weight of the observation at grid location *j* for estimating the regression coefficient at grid location *i*

$d_{ij}$ = Euclidean distance between *i* and *j*

$\theta_{i(k)}$ = adaptive bandwidth size of *k*-th nearest neighbor distance.

All parameter estimations were performed using GWR v4.0.

### 2.4. Spatial Downscaling of Annual TRMM Data

The RF and MGWR models were tested to develop the associations between TRMM precipitation data and explanatory variables (i.e., elevation, slope, normalized difference vegetation index (NDVI), land surface temperature (LST), actual evapotranspiration (AET), wind speed (WDS), and cloud cover) at coarse spatial resolution. Models developed at coarse resolution were further used with high-resolution explanatory variables to forecast the high-resolution precipitation estimates. The downscaling process of the TRMM precipitation is explained in detail in Figure 2 and follows these steps:

1.  All high-resolution explanatory variables (1 km × 1 km) were resampled to the TRMM grid scale (0.25° × 0.25°). The resampled explanatory variables and TRMM precipitation were used as an input in the RF and MGWR models to predict the coarse resolution (0.25° × 0.25°) precipitation estimates (step 1 in the flow chart).
2.  The estimated precipitation (0.25° × 0.25°) was subtracted from TRMM observations (0.25° × 0.25°) to calculate the residuals (error) in the model's estimates.
3.  The performance evaluation is conducted to select the appropriate model from RF and MGWR (step 1 in the flow chart).
4.  The model regression coefficients from step 1 were interpolated to high resolution (1 km × 1 km). The high-resolution explanatory variables (1 km × 1 km) and regression coefficients were used in Equation (2) to obtain the fine scale (1 km × 1 km) estimated precipitation (step 2 in the flow chart).
5.  Finally, residuals from step 1 were interpolated to high resolution (1 km × 1 km) and subsequently combined with the estimated precipitation to obtain the downscaled-TRMM precipitation estimates (step 2 in the flow chart).

### 2.5. Combining Annual Downscaled TRMM Data with APHRODITE

The annual downscaled precipitation data are combined with APHRODITE rain gauge-based precipitation data using the spatial GRA (Geographical ratio analysis) algorithm using the following expressions (step 3 in the flow chart).

$$P_{cal_{downscaled}}(y) = P_{downscaled}(y) \times \sum_{j=1}^{n} \times \lambda_j \frac{P_{APHRODITE} \, Y_j}{P_{downscaled} \, Y_j} \tag{5}$$

where

$P_{cal_{downscaled}}(y)$ = merged or calibrated-downscaled precipitation with APHRODITE rain gauge-based data at 1 km grid "y"

$P_{APHRODITE} \, Y_j$ = APHRODITE rain gauge-based precipitation at APHRODITE grid "*Y*"

$P_{downscaled} \, Y_j$ = downscaled precipitation values at APHRODITE grid "*Y*"

$\lambda_j$ = Spatial weights at each grid location $(Y)_j$

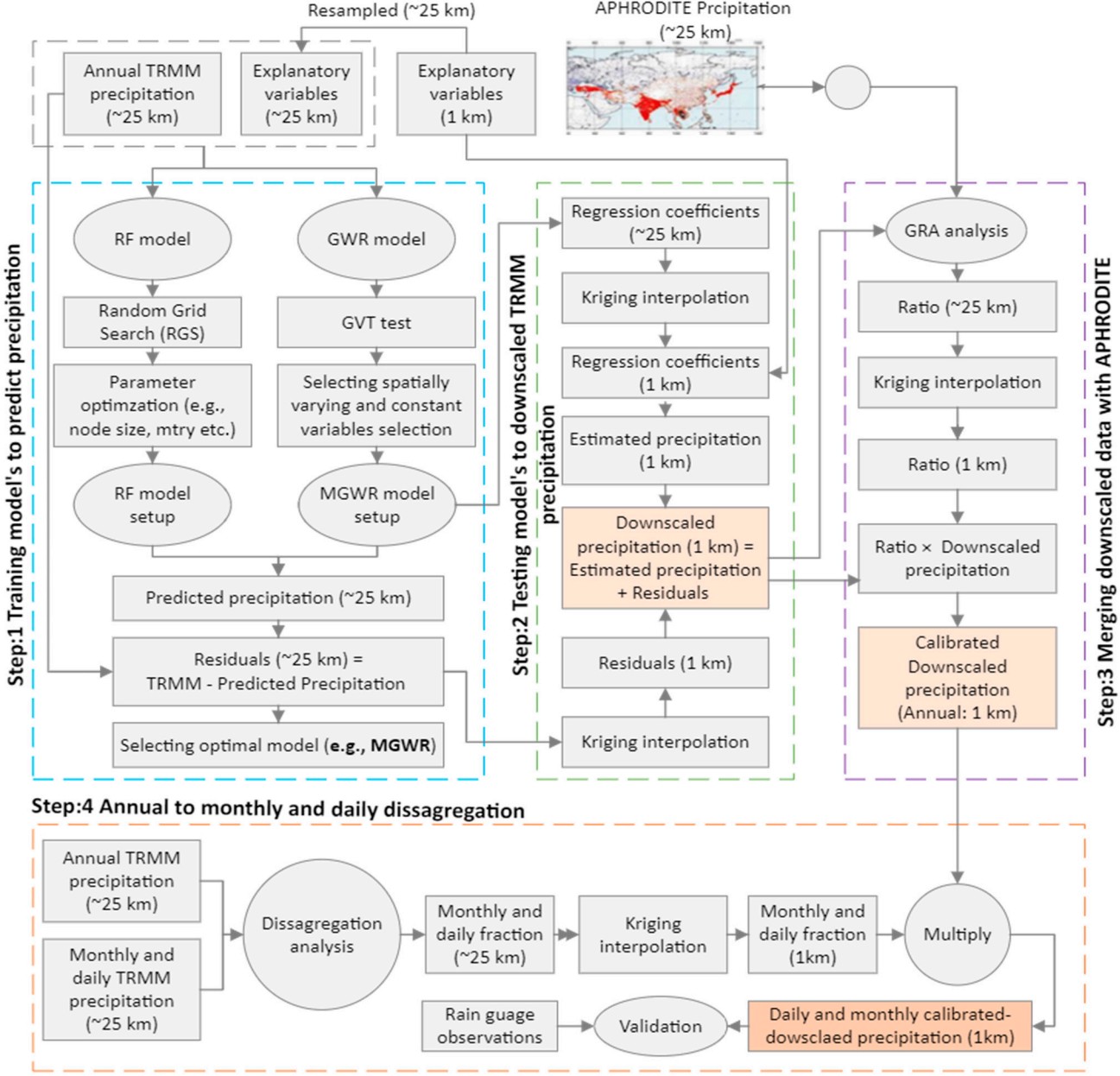

**Figure 2.** Flow chart indicating data collection and downscaling framework.

### 2.6. Monthly and Daily Downscaled Precipitation Estimates

A fractional disaggregation approach [45,58] was utilized to disaggregate the annual high-resolution (1 km) downscaled calibrated precipitation data to acquire the monthly and daily precipitation estimates using the following expression (step 4 in the flow chart):

$$TRMM_{downscaled} \, (monthly \ and \ daily)^{1km} = Annual \ TRMM_{downscaled}(1km) \times \sum_{i=1}^{N} \lambda_i \left( \frac{TRMM_j \, (x_i)^{0.25 \ deg}}{\sum_{j=1}^{n} TRMM \, (x_i)^{0.25 \ deg}} \right) \quad (6)$$

where:

$TRMM_j \, (x_i)^{0.25 \ deg}$ = coarse resolution TRMM precipitation at grid location $(x_i)$ of the jth day) $\sum_{j=1}^{n} TRMM \, (x_i)^{0.25 \ deg}$ = annual precipitation for $n$ = 12 months and 365 days

$\lambda_i$ = weight at each grid location $x_i$ with a spatial resolution of $0.25° \times 0.25°$ and $N$ represents the total number of grids in the study region.

### *2.7. Evaluation Matrices*

The downscaled and calibrated results of TRMM precipitation estimates were compared with rain gauge precipitation to evaluate the performance using several different statistical indicators, e.g., coefficient of determination ($R^2$), Mean Square Error (MSE), Bias, Root mean square error, and (RMSE) (Table 2).

## 3. Results

### *3.1. Comparison between MGWR and RF-Based Estimated Precipitation*

MGWR and RF models were trained with coarse-resolution environmental variables and TRMM precipitation data. The estimated annual precipitation results from RF and MGWR models were compared with the coarse-resolution TRMM precipitation data. Figure 3a,b indicate scatter plots between the estimated and TRMM precipitation from 2000–2019. Results indicated that the MGWR model fitted better than the RF with higher values of $R^2$ as 0.96. In the case of the RF model, the points are more scattered around the 1:1 ratio line, which seems to be a better fit with higher accuracy in the case of the MGWR model. Figure 3c illustrates the residual distribution of both models based on the probability distribution function (PDF). The line graph skewed on the right side and left side from 0 means demonstrated that the model's estimated precipitation is over- and under-estimated compared to the original TRMM data. The residual values ranged between 500 and −500 mm/year. The PDF plots also confirmed that the MGWR model had lower residuals (error) than the RF model, as evidenced by the fact that most error values were distributed around zero with a higher peak density, which seems to be found with a higher spread in the case of the RF model. The time-series estimated precipitation data from both models were also compared for selected grids across upstream and downstream of the study region (Figure 3d). It is observed that a good agreement is found between time-series data of MGWR-based estimated precipitation and TRMM precipitation estimated through the study period from 2000–2019 over all the selected grids. In contrast, a weaker agreement is observed in RF-based estimated precipitation.

The spatial patterns of estimated precipitation obtained from the RF and MGWR models were also compared with the original TRMM precipitation for the average study period (2000–2019) (Figure 4a–c). It is well noted that MGWR-based estimated precipitation shows a similar spatial pattern compared to the original TRMM precipitation data. The negative values of residuals for the RF model were found higher in the upstream regions, indicating that the RF model does not perform satisfactorily and underestimates the precipitation in the regions with higher precipitation values (Figure 4d). On the other hand, over- and under-estimations of MGWR-based precipitation results were relatively lower and showed a homogeneous pattern in the study region, indicating that MGWR model performance is highly satisfactory in predicting the precipitation data (Figure 4e).

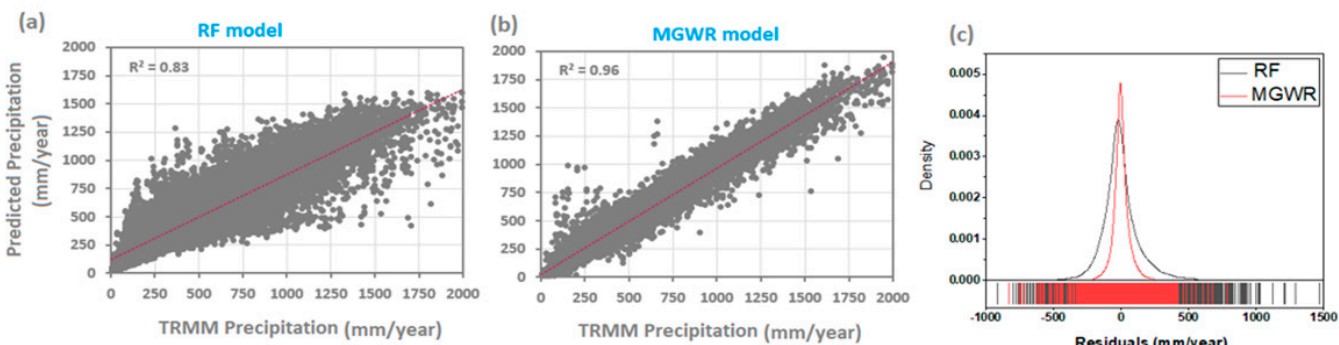

**Figure 3.** *Cont.*

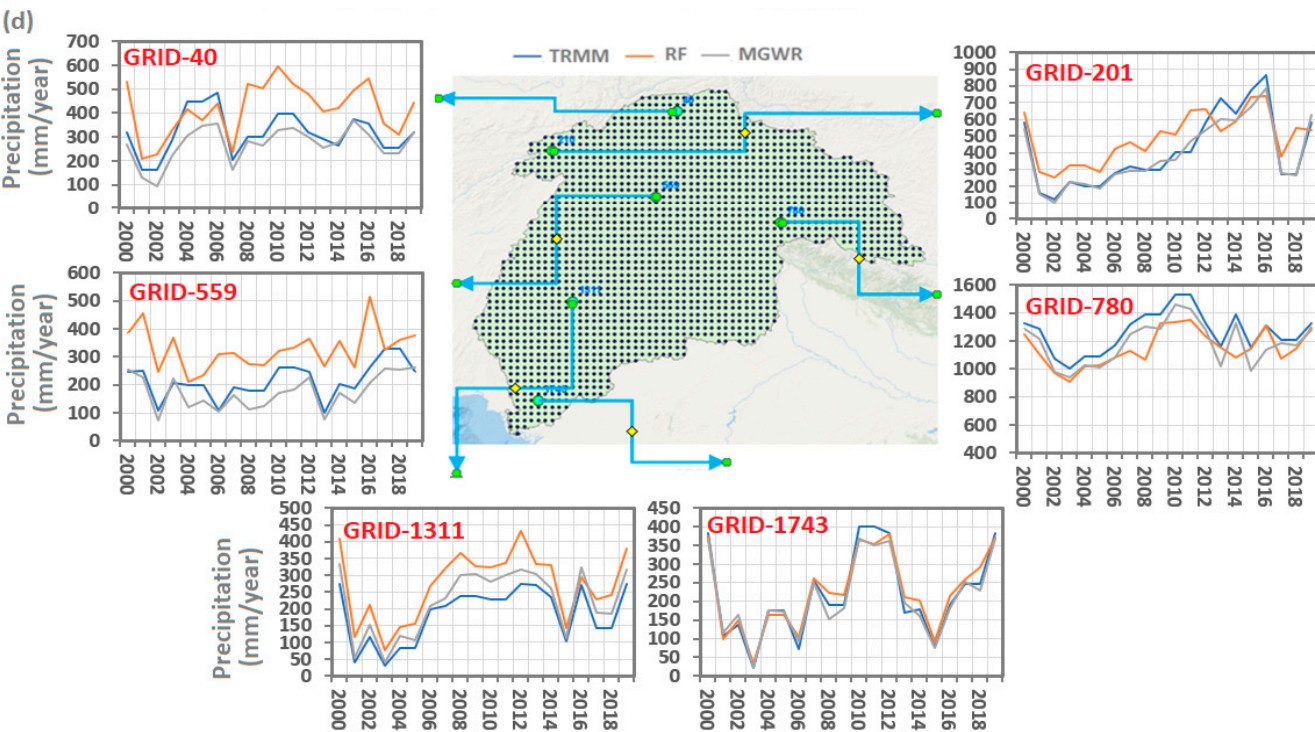

**Figure 3.** (**a**,**b**) Comparison between the TRMM precipitation and RF model estimated and MGWR model estimated precipitation, (**c**) probability distribution function (PDF) of model's residuals, and (**d**) time-series comparison between the TRMM precipitation and model's estimated precipitation over different selected grids in the Indus Basin from 2000 to 2019.

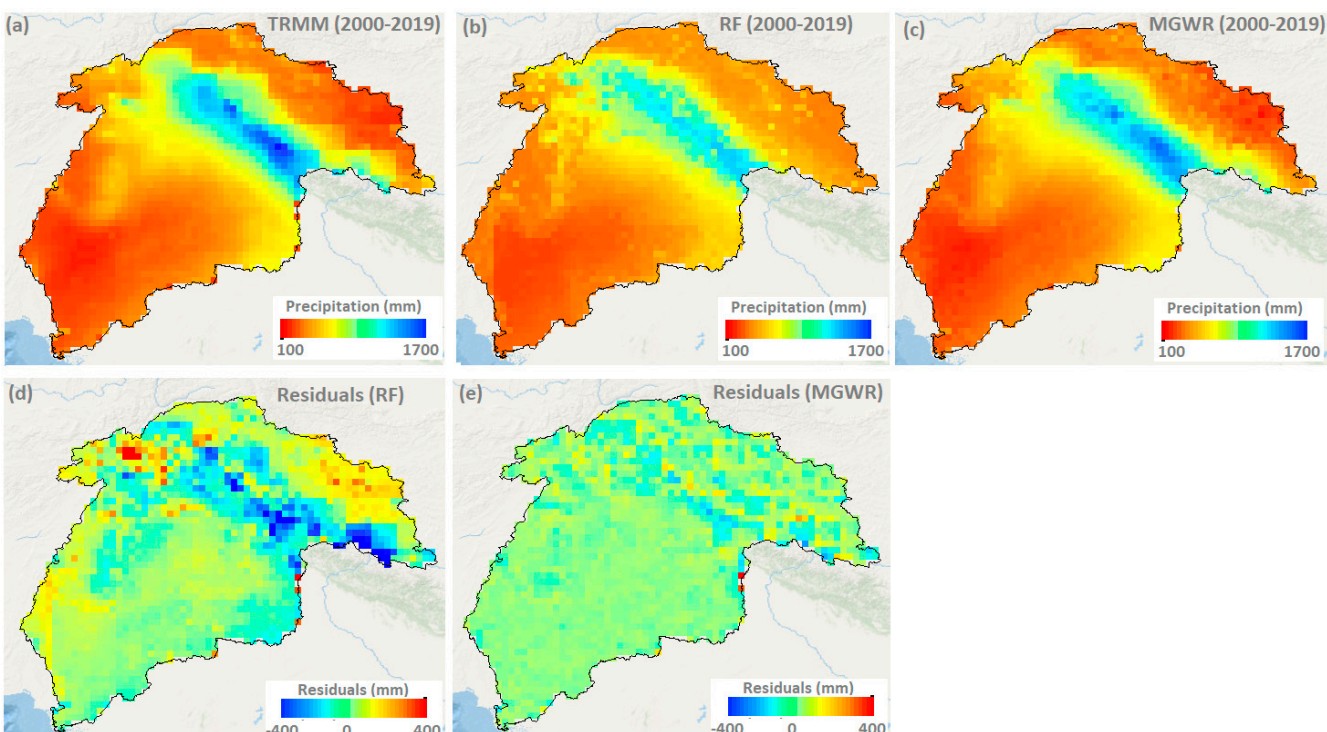

**Figure 4.** Comparison between the spatial pattern of average precipitation (2000–2019) obtained from the original TRMM data, RF model, and MGWR model (**a**–**c**) and spatial pattern of residuals obtained from the RF and MGWR models (**d**,**e**) at ~25 km resolution.

Further detailed yearly evaluations of the model's performances based on different statistical indicators ($R^2$, MSE, and mean) are provided in Figure 5a–c. Based on $R^2$ values, the MGWR model performance was satisfactory through the study period from 2000 to 2019, with higher values of $R^2$ ranging between 0.97 and 0.99. On the other hand, the $R^2$ values of the RF model ranged between 0.88 and 0.93 (Figure 5a). The MSE values of the MGWR model were comparatively less (ranging from 38 to 79 mm/year), which seems to be higher in the case of RF model (120 to 163 mm/year) (Figure 5b). Overall, the detailed statistical evaluation indicated that the MGWR model fit better with lower MSE and with higher values of $R^2$ and outperformed the RF model to predict the precipitation based on current settings of environmental variables.

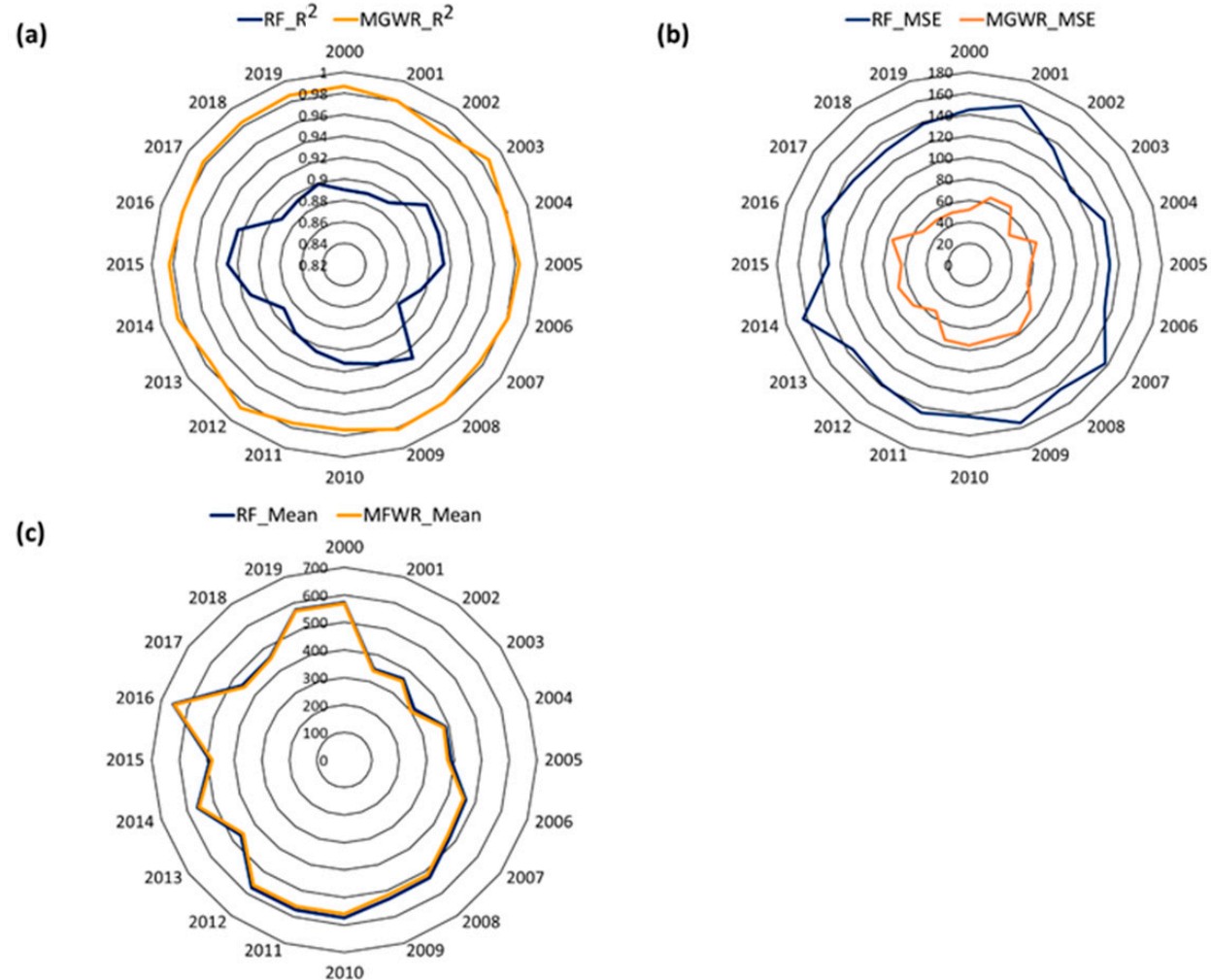

**Figure 5.** Performance assessment of RF and MGWR models results based on the (**a**) $R^2$, (**b**) MSE, and (**c**) mean in the estimated precipitation from 2000 to 2019.

*3.2. Spatial Downscaling of Annual TRMM Precipitation*

We ran the MGWR model with annual TRMM precipitation in combination with coarse resolution (~25 km) explanatory variables (wind speed, cloud cover, elevation, slope, ET, NDVI, and LST) to obtain the model's regression coefficients (see Figure S3 in the Supplementary Material). The regression coefficients of explanatory variables obtained at this stage were subsequently interpolated to high resolution (1 km) and integrated with high-resolution environmental variables to obtain the estimated precipitation at a 1 km resolution (Figure 6b). Further, residual correction is performed by adding estimated precipitation with model residuals to obtain the high-resolution TRMM downscaled data

(Figure 6b,c). It is observed that the downscaled TRMM precipitation data shows a significant improvement in the spatial information (Figure 6c). Further, the precipitation profile along the ~1200 km transect line shows a very strong resemblance in the precipitation pattern obtained from both datasets. However, precipitation variations along the ~1200 km transect line obtained from the downscaled TRMM data translate a more variable trend compared to the coarse resolution TRMM data (Figure 6d).

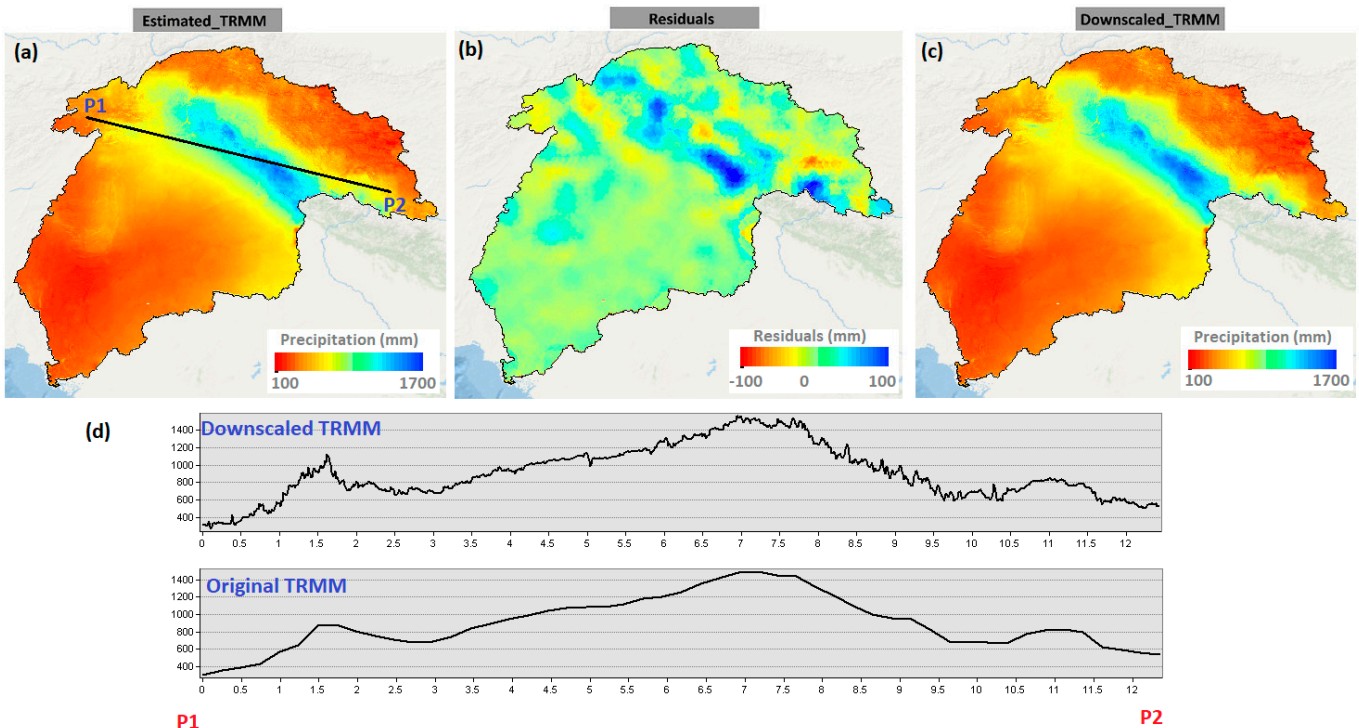

**Figure 6.** Output maps of MGWR model indicating the (**a**) estimated precipitation (1 km), (**b**) residuals (1 km), (**c**) downscaled TRMM precipitation (1 km), and (**d**) precipitation pattern obtained from the downscaled TRMM and original TRMM along the ~1200 km line of the transect. (Note: data presented here are average of 2000–2019).

MGWR performance is tested for each year from 2000 to 2019 to generate high-resolution (1 km) precipitation data for the Indus Basin. Figure S4 compares the spatial pattern of annual precipitation estimates obtained from the downscaled TRMM data and original TRMM data for different selected years from 2000 to 2019. It is observed that downscaled precipitation estimates indicated quite similar spatial patterns of precipitation values when compared with the original TRMM data for each year. This indicated that the downscaling approach with current high-resolution explanatory variables not only translates precipitation with enhanced information but also retains the spatial pattern of precipitation throughout the study period. Here, we applied the downscaling approach over the original TRMM data; any bias that was already present in the satellite data will eventually be introduced in the model-based downscaled precipitation results [45,67]. Therefore, we combined the APHRODITE rain gauge-based precipitation data with downscaled TRMM results to remove the uncertainties in the downscaling results and improve the precision of precipitation estimates. Figure 7 represents the calibrated results of precipitation estimates and their comparison with original TRMM and downscaled TRMM data for three selected years, 2007, 2009, and 2011. We can see that the calibrated TRMM precipitation results differed from the downscaled and original TRMM data (Figure 7). The difference was more prominent across the upstream regions, indicating that uncertainties in the downscaled results were higher in these regions and the current merging approach helps reduce the uncertainties in the final precipitation estimates.

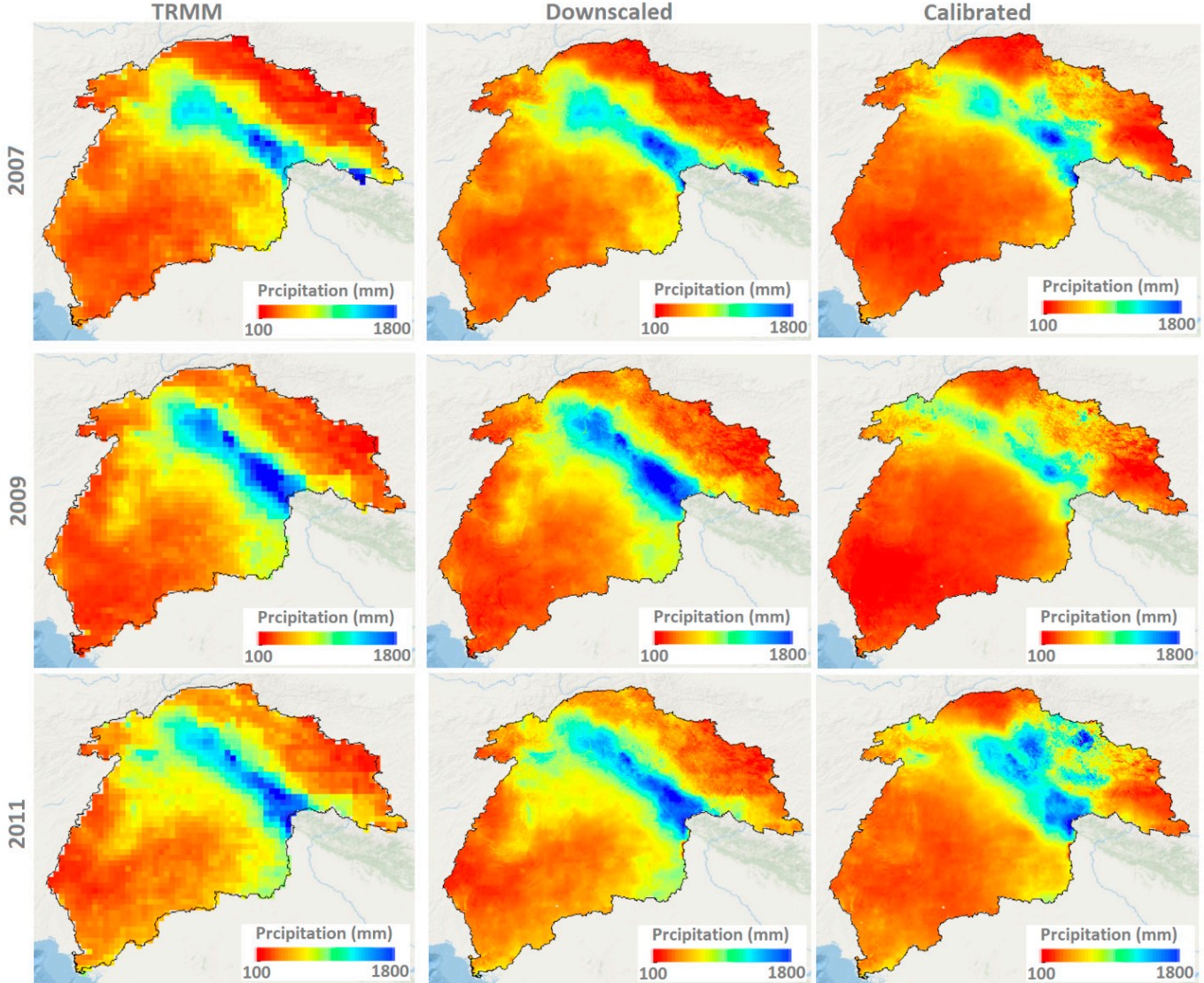

**Figure 7.** Spatial pattern of coarse resolution TRMM, downscaled TRMM, and calibrated TRMM precipitation results estimated for three selected years.

Further, statistical matrices and scatter plots show performances of annual TRMM, downscaled, and calibrated precipitation estimates corresponding rain gauge data across the (a) elevation zone 1, (b) elevation zone 2, (c) elevation zone 3, and (d) average of the entire study Basin (Figure 8). Results indicated that TRMM results after downscaling (represented as 'downscaled') and subsequently merged with APHRODITE (represented as 'calibrated') show an improvement in precipitation estimates regarding rain gauge data across all elevation zones. For instance, $R^2$ improved from 0.67 to 0.76 with a reduction in bias from 29.21% to −1.13% and RMSE from 210.48 mm/year to 133.73 mm/year across the elevation zone 1 (0–500 m). Over the entire basin, the precipitation estimates after downscaling and calibration were improved with a reduction in RMSE and bias values by 222.25 mm/year to 134.79 mm/year and 29.01% to −1.96%, respectively.

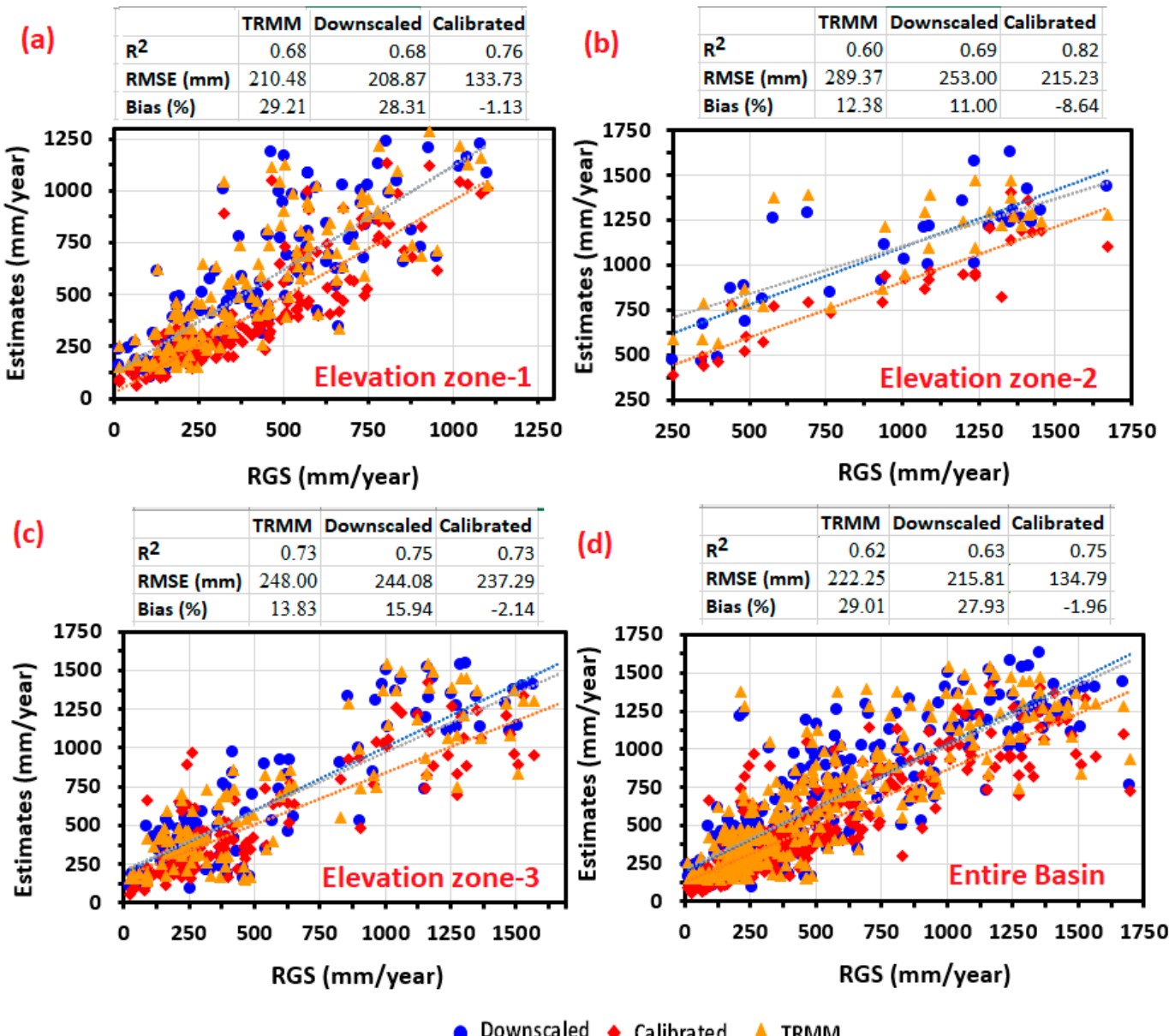

**Figure 8.** Statistical evaluation matrices and scatter plots showing performances of TRMM, downscaled, and calibrated precipitation estimates compared to rain gauge data (RGS) across the (**a**) elevation zone 1, (**b**) elevation zone 2, (**c**) elevation zone 3, and (**d**) average of entire study Basin.

### 3.3. Temporal Disaggregation of Annual Downscaled TRMM Data

The downscaled and calibrated precipitation results of annual TRMM data were temporally downscaled (or disaggregated) to acquire the precipitation estimates for the monthly and daily temporal scales. The monthly precipitation estimates for Jan, May, and September of the year 2011 obtained after disaggregation of the downscaled TRMM (1 km) show similar spatial patterns when compared with the coarse resolution TRMM results (~25 km) (Figure 9). The monthly precipitation results from the coarse resolution TRMM displayed a mosaic-like information because of the coarse spatial scale of TRMM data. Precipitation obtained from a MGWR-based downscaling approach, on the other hand, not only retained the spatial pattern but also enhanced spatial information (see more images in Figure S5 in the Supplementary Material). Monthly precipitation results from the calibrated-downscaled data resulted in a significant difference in precipitation estimates between the downscaled and calibrated-downscaled, notably in the high-altitude

mountain regions (Figure 9). After merging with the APHRODITE data, the adjusted amount of precipitation is responsible for the significant disparity between downscaled and calibrated-downscaled precipitation estimations.

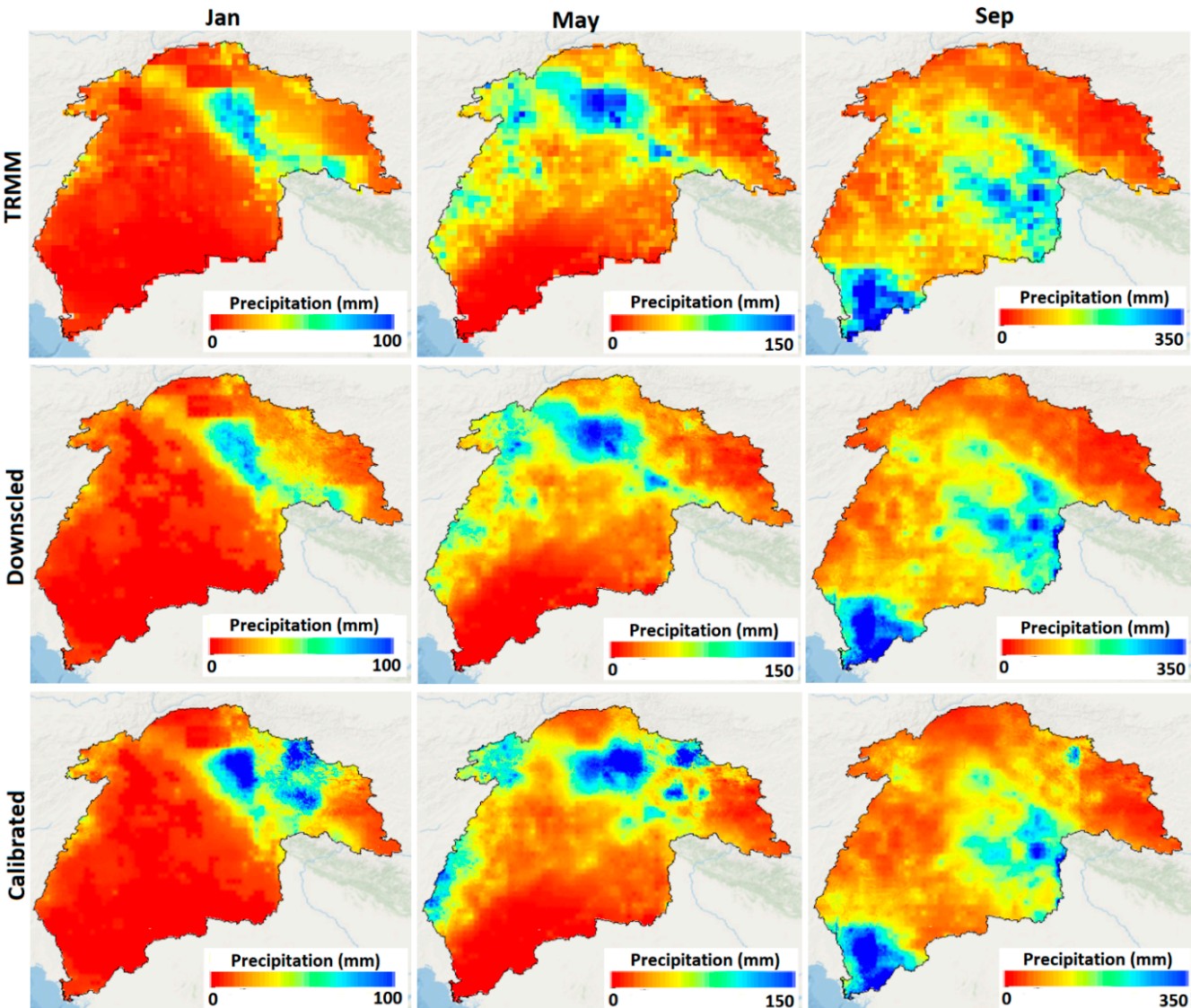

**Figure 9.** Spatial pattern of precipitation results obtained from the coarse resolution TRMM, downscaled TRMM, and calibrated TRMM precipitation at monthly scale for January, May, and September of the year 2011.

Results of the spatial pattern of daily precipitation estimates disaggregated from the coarse resolution TRMM, downscaled TRMM, and calibrated downscaled data for different days of the year, e.g., 1, 32, 60, 91, 121, 152, 182,213, 244, 274, 305, and 356, are shown in Figure 10. It is observed that the daily precipitation estimates obtained from downscaled and calibrated-downscaled datasets also offered better spatial information when compared with the coarse resolution TRMM data. Remarkably, daily precipitation results obtained from the calibrated dataset had a significantly different spatial pattern compared to those produced from the downscaled dataset. However, in most of the study areas, downscaled and calibrated-downscaled precipitation patterns were similar with some variations. These facts demonstrated that the temporal disaggregation of annual precipitation to monthly and daily scales does not affect the results to a higher extent.

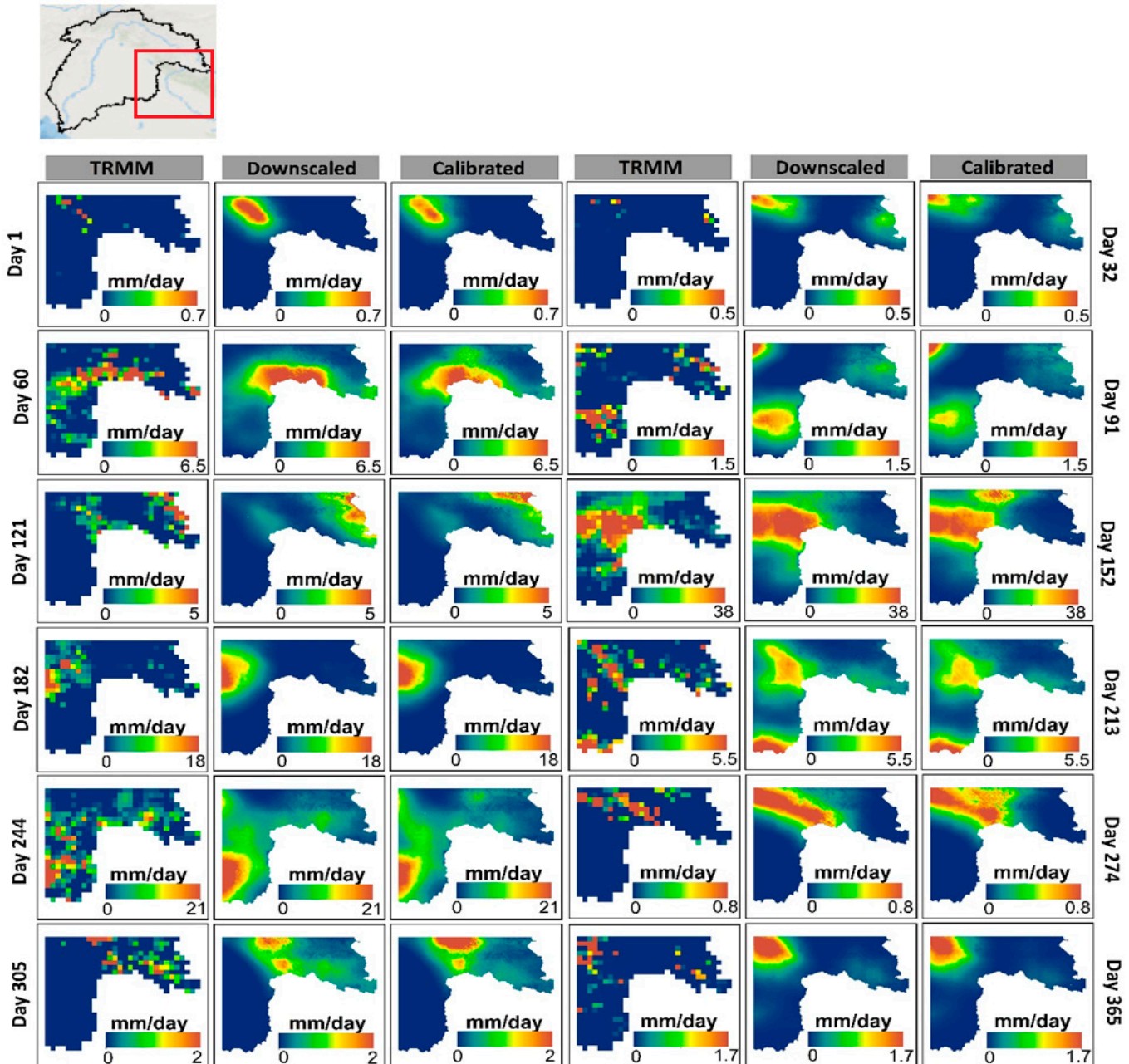

**Figure 10.** Snap view of the spatial pattern of daily precipitation estimates from coarse resolution TRMM, downscaled, and calibrated datasets for different days of the year such as 1, 32, 60, 91, 121, 152, 182,213, 244, 274, 305, and 356. Red box indicated the area of interest in the Indus Basin.

We compared the results with rain gauge data to confirm the accuracy of the monthly and daily precipitation estimates disaggregated from downscaled and calibrated-downscaled data. It is noted that a good agreement is found between the regional times series of estimated precipitation (TRMM, downscaled, and calibrated) and rain gauge-based precipitation data (Figure 11a). The performance evaluation over the entire basin indicated that precipitation estimates of TRMM data after downscaling and calibration procedure show a slight increase in $R^2$ from 0.79 to 0.80. However, a significant improvement was observed with a higher reduction in RMSE and bias in the TRMM precipitation data as 43.02 mm/month to 14.16 mm/month and 15.73% to 5.25% after downscaling and calibration procedures (Figure 11b). In general, the correlation coefficient between the estimated

precipitation and rain gauges was comparatively higher across elevation zone 1 and zone 2 (Figure 11c,d) than in elevation zone 3 (Figure 11e).

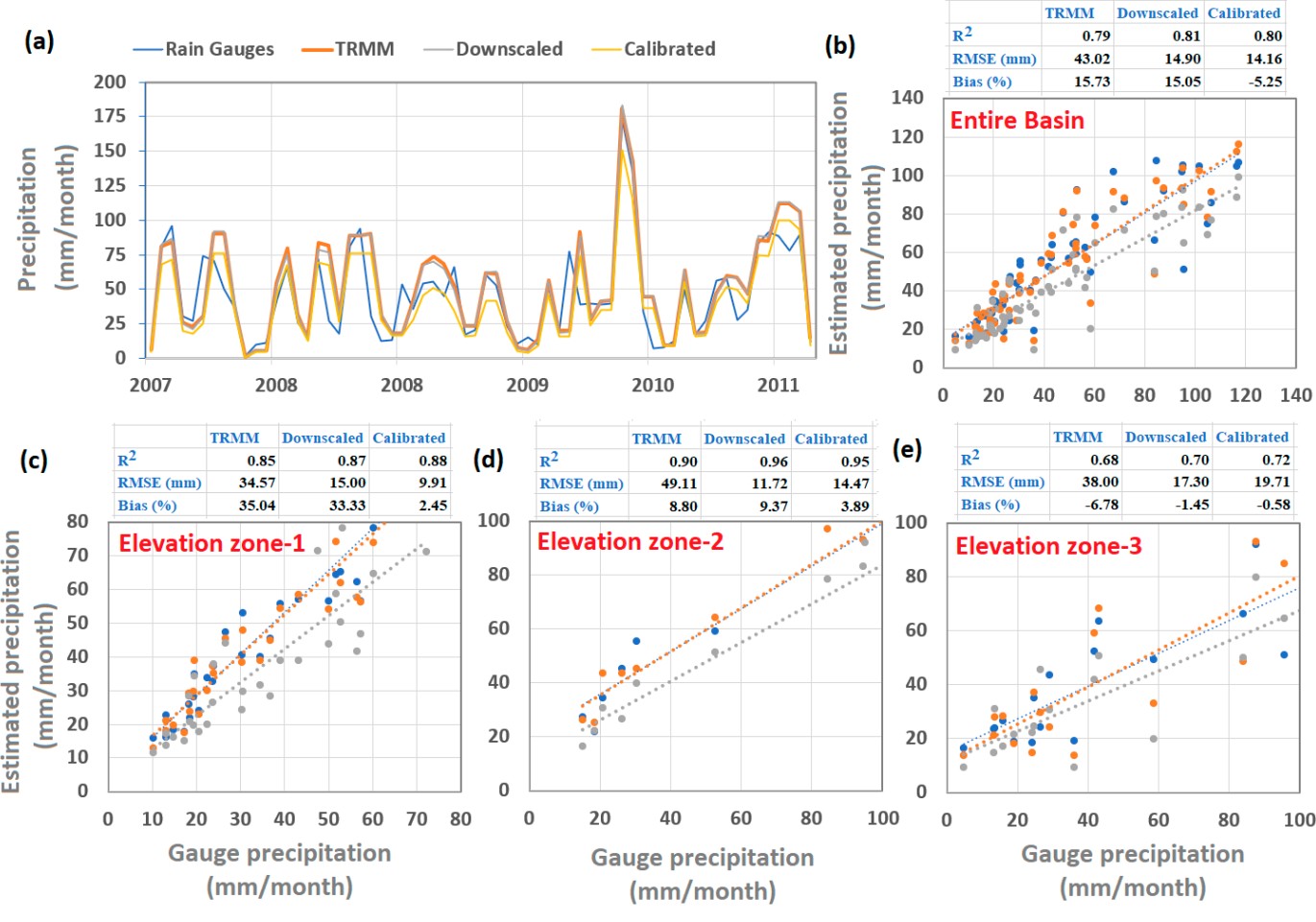

**Figure 11.** (**a**) Comparison between monthly time series of precipitation estimates (coarse resolution TRMM, downscaled, and calibrated results) with rain gauge data, and (**b–e**) scatter plot showing the performance evaluation of different precipitation estimates compared to rain gauge data across three elevation zones and the entire basin.

We further explored the performances of the daily precipitation by comparing results with rain gauge observations. It is observed that downscaled and calibrated results of daily precipitation show strong agreement when compared with daily time series of rain gauge data (Figure 12a). Furthermore, the performance of precipitation estimates from TRMM, downscaled, and calibrated datasets is evaluated using different statistical indicators (correlation coefficient, root mean square error, and bias) across the entire Basin and selected stations across the three elevations zones (Figure 12b–e). For instance, performance evaluation over the entire Basin indicated that TRMM precipitation estimates showed an improvement with a decrease in RMSE and Bias values from 1.42 to 0.50 mm/day and 7.25% to 3.32%, respectively, and an increase in $R^2$ from 0.82 to 0.85 (Figure 12b). Based on bias values, TRMM-based precipitation estimates have relatively higher over- and underestimations across all elevation zones, which seems to be lower in downscaled and calibrated results, suggesting that the present downscaling-calibration measures advance the precision in daily precipitation estimates (Figure 12c–e). In general, the higher reduction in bias values after downscaling and calibration procedures was noted across the downstream low-elevation zones (e.g., zone 1 correspond to elevation changes from 0 to 500 m).

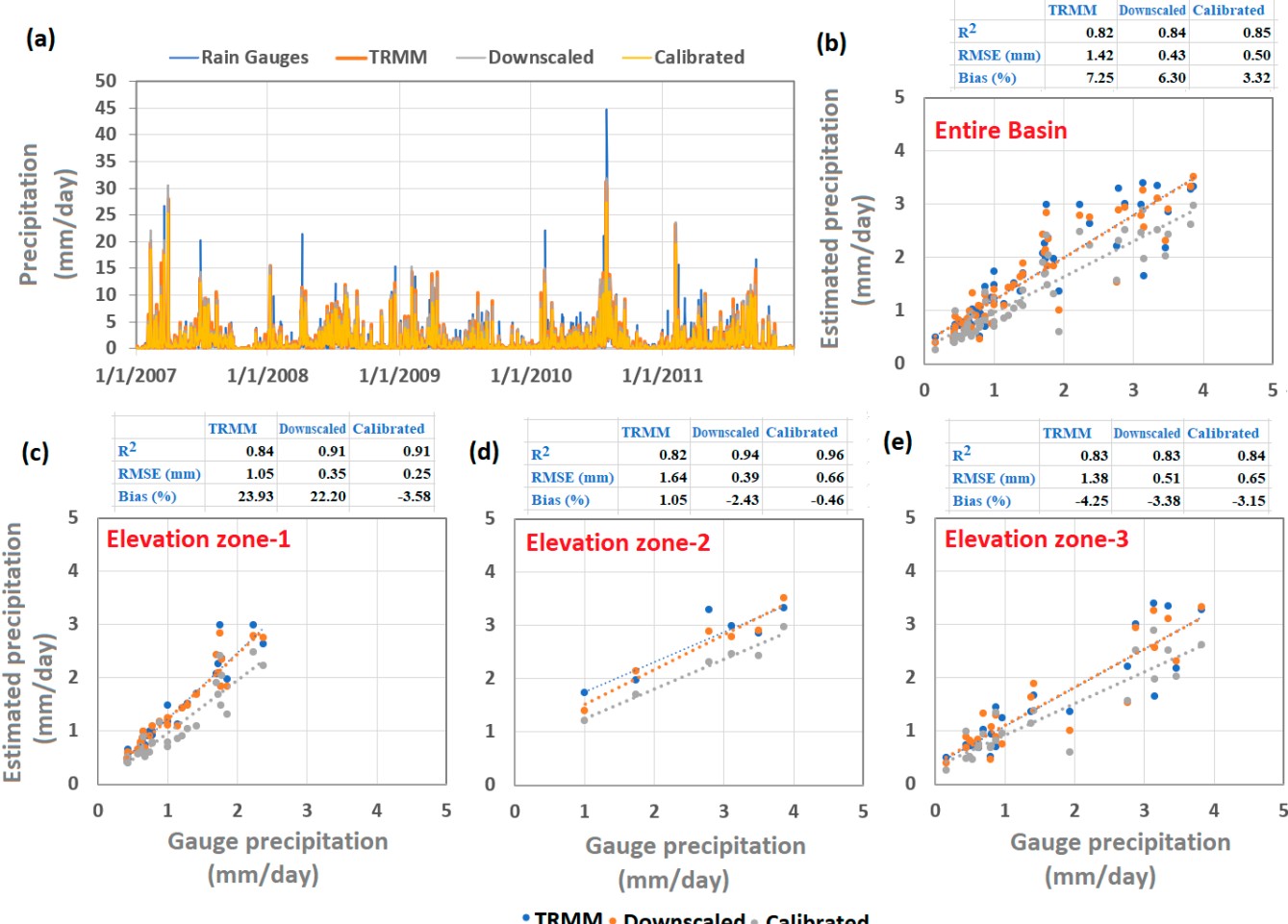

**Figure 12.** (**a**) Comparison between daily time series of precipitation estimates (TRMM, downscaled, and calibrated) with rain gauge data, and (**b**–**e**) scatter plot showing the performance evaluation of different precipitation estimates compared to rain gauge data across three elevation zones and an entire basin.

## 4. Discussion

The current study reconstructed the high-resolution precipitation forecast by combining APHRODITE rain gauge-based precipitation with downscaled TRMM data. The estimated precipitation forecast performance was also investigated across the Indus Basin in different elevation zones. Here, we discussed the following key points.

The selection of appropriate explanatory variables plays a critical role in the downscaling procedure [45,68]. The seven high-resolution explanatory variables explaining the physiographical and hydro-climatic conditions of the Indus Basin were selected, including elevation, evapotranspiration, slope, vegetation characteristics, land surface temperature, wind speed, and cloud cover conditions. Variables were chosen because of their regional significance [69] and overall effects on the changing trend in precipitation patterns [45,70]. Other studies have well-reported applications of these explanatory variables over the other global watersheds [55,71–73]. The results of the variable importance test (VIMP) to evaluate the relative importance of each explanatory variable indicated that wind speed was found to be of relatively high importance (58%), followed by the NDVI (47%) and elevation (47%) (Figure 13). The strong association between precipitation and NDVI and elevation has also been reported by [74]. The actual ET, LST, and slope were found to be equally important in the model for predicting precipitation data. Several variables were introduced in the RF and MGWR model to predict the high-resolution precipitation forecast in the Indus Basin,

although the geographically weightage regression (GWR) model has a widely applied approach to building non-stationary relationships between the explanatory variables and prediction (TRMM in our case) [45,75,76], and these relationships facilitate the downscaling spatial resolution of gridded satellite products [45,74]. The GWR model considers all input variables' coefficients as non-stationary, which is invalid. The coefficients of some variables could be spatial varying (non-stationary) and fixed (constant or stationary) for some variables [45,74,77]. Introducing all variables' coefficients as non-stationary reduces the model fitting and could result in uncertainties [45,49,50]. Therefore, detailed characterization of explanatory variables and proper selection plays a vital role in model fitting. In our study, we separated the stationary and non-stationary coefficients based on the GVT test to build an MGWR model (see Table S1 in the Supplementary Material). This model handles both types of coefficients to increase the model fitting. Previous researchers also proved that the MGWR model outperformed the GWR model when appropriate variables based on their stationary and non-stationary behaviors were introduced in the models [45,50,77,78]. Therefore, we introduced the applications of the MGWR model to downscale TRMM, and its performance was also compared with a machine learning (RF) model. We found that the MGWR model outperformed the RF model, which has also been reported in previous studies [46,79,80].

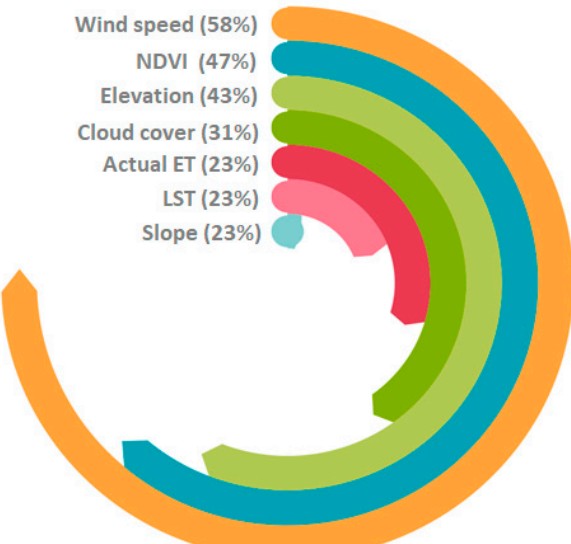

**Figure 13.** Regional feature importance of different explanatory variables for precipitation prediction in the study region.

The downscaling of TRMM precipitation data to fine sale improved the spatial information and reduced the bias in data compared with rain-gauge values. At the same time, a higher inconsistency is observed between the coarse-resolution TRMM precipitation and rain-gauge observations. This inconsistency may be due to coarse pixel resolution (0.250 × 0.250), which may include a mixed nature of rain and non-rain zones and cause uncertainty in the results [81,82]. Precipitation values could deviate by ~38% between two gauges located within a single satellite pixel (4 km × 4 km) [56,81] also reported that the mean deviation of rain gauge observations and TRMM precipitation over the single pixel was relatively higher than when calculated with a cluster of pixels at finer scales. To further prove this evidence, we compared the precipitation estimates acquired from the coarse resolution TRMM and downscaled data over the single-pixel (25 km × 25 km) corresponding to two rain gauges (Lahore (AP) and Lahore (PBO)) (Figure 13). It is well noted that downscaled TRMM estimates show a similar spatial precipitation pattern compared to the original TRMM data. However, downscaled data well translated the spatial heterogeneity in the precipitation values across the area of interest, which seems to be

homogeneous in the case of the original TRMM. We see that two rain gauges located in a single pixel at ~108 km show a difference in observed precipitation of 48 mm, indicating that precipitation varies at a smaller scale which cannot be translated in the single satellite pixel of TRMM (e.g., indicating precipitation of 685 mm/year through the selected region). On the other hand, downscaled-based precipitation showed a significant spatial change even within a single pixel and translated the changes corresponding to the rain gauge values. The deviations of TRMM precipitation from gauge observations were recorded as 73 mm and 121 mm, corresponding to the Lahore (PBO) and Lahore (AP) gauge stations, respectively. In comparison, the deviations of downscaled precipitation estimates were 66 mm and 14 mm for Lahore (PBO) and Lahore (AP) gauge stations, respectively (Figure 14). The larger deviations between the single rain gauge observation and TRMM precipitation records are mainly associated with the scale of discrepancy [82]. It could be decreased either by improving the calibration process with dense rain gauge networks or rescaling the satellite to a high spatial resolution [83], which also supports our findings.

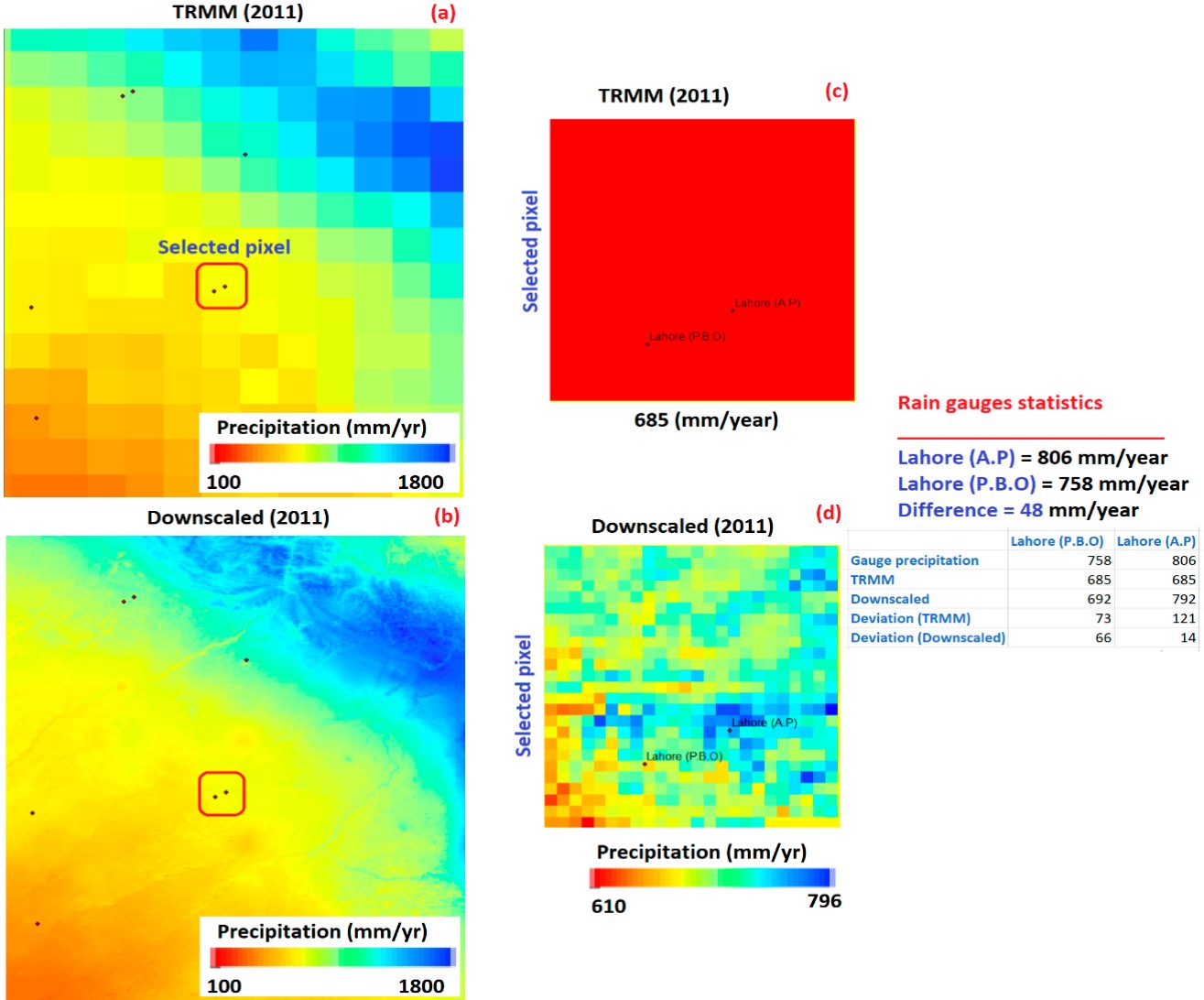

**Rain gauges statistics**

Lahore (A.P) = 806 mm/year
Lahore (P.B.O) = 758 mm/year
Difference = 48 mm/year

|  | Lahore (P.B.O) | Lahore (A.P) |
|---|---|---|
| Gauge precipitation | 758 | 806 |
| TRMM | 685 | 685 |
| Downscaled | 692 | 792 |
| Deviation (TRMM) | 73 | 121 |
| Deviation (Downscaled) | 66 | 14 |

**Figure 14.** Spatial view of (**a**) coarse resolution TRMM, (**b**) downscaled TRMM (1 km) for the year 2011, (**c**) snap view of TRMM precipitation over a single pixel (25 km × 25 km), and (**d**) corresponding snap view of downscaled precipitation over single pixel (25 km × 25 km).

Downscaled results still contain higher values of bias, which might be associated with the downscaling procedure applied to original TRMM data, and any errors associated with

satellite products could be induced in the final downscaled results. Therefore, additional preprocessing is required to improve the precision of downscaled precipitation estimates. Merging rain-gauge data with satellite estimates provides a favorable solution to enhance the precision of precipitation estimates through calibration techniques [37,45,56,84]. Since the current study covers the entire Indus Basin and the sparse distribution of gauge networks, especially in the upstream high-altitude areas [63,85], restricts the potential of merging and calibration methods to improve the reliability of precipitation, therefore, the APHRODITE dataset was used in the merging process, and available gauges were only used to assess the performance of TRMM, downscaled, and calibrated downscaled (or merged) products. In the present study, we used the GRA approach for merging TRMM data with APHRIDTE as recommended by [45,56]. Our findings indicated that the accuracy of TRMM precipitation estimates greatly improved when we [86,87] merged with APHRODITE rain gauge-based data. [88] also reported that the dense rain gauge network dataset (APHRODITE) significantly enhances the accuracy of TRMM after merging. GRA and GDA approaches are simple and effective techniques to adjust over- and underestimation in the model-based downscaled and satellite estimates

We evaluated the quality of precipitation estimates acquired from the downscaled and calibrated results in reference to the original TRMM estimates across three different elevation zones, e.g., 0–500 m, 500–1000 m, and >1000 m. We found that the TRMM precipitation forecast after downscaling and calibration procedures showed a greater reduction in error across the low-altitude regions (e.g., zone 1 and zone 2) than the high-altitude regions. In contrast, the performances of TRMM estimates were slightly lower across the high-altitude regions (e.g., zone 3). The low performance of precipitation products across the elevation zone 3 might be associated with the fact that these regions are covered with glacier and seasonal snow, and satellite observations are subjected to uncertainties [45,63,89]. The annual, monthly, and daily downscaled results of the precipitation forecast were still satisfactory, given higher values of $R^2$ and lower bias compared with available ground observations data. The current study is mainly designed to improve the spatial (horizontal) information of TRMM precipitation across the Indus Basin; however, further efforts are required to adjust the vertical amount of precipitation, mainly in the high-altitude mountain regions with glacier coverage, which consists of sparse rain gauges, and the precipitation measurements are unreliable or unavailable [63,90].

## 5. Conclusions

This study developed a downscaling-calibration approach to translating high-resolution spatio-temporal precipitation estimates by combining downscaled TRMM with APHRODITE rain gauge-based data in the Indus Basin. We compared the performances of MGWR and RF models in the downscaling framework. The downscaled precipitation estimates were also merged with the APHRODITE data using the GRA approach to enhance the accuracy. The downscaled and calibration results generated by this study were compared with the rain-gauge observations for different elevation zones to ensure the accuracy of the results. Overall, the main findings of this study include:

- The GVT test indicated that LST and slope coefficients are stationary and hence should be switched in the global part of the model, and the rest of the variables were introduced as local variables with non-stationary behaviors.
- The MGWR model performed better with higher fitting and accuracy to predict the precipitation than the RF model.
- Downscaled TRMM datasets not only translate the spatial heterogeneity of precipitation estimates but also reduce the deviation in results when compared with rain gauge observations.
- The accuracy of the downscaling results was considerably increased after combining them with APHRODITE rain gauge-based precipitation data.

- Evaluation across different elevation zones indicated that the downscaling-calibrated procedure improved the quality of precipitation estimates across the low-elevation zones, while a slight improvement was observed across the high-altitude regions.

High-resolution data generated by the proposed MGWR-based modeling framework would assist water resource managers to use it for the decision making process at distributed basis, particularly over the high-altitude areas, which have the sparse distribution of gauge observations. The method can also be encouraged to test over similar regions of other global watersheds.

**Supplementary Materials:** The following supporting information can be downloaded at: https://www.mdpi.com/article/10.3390/rs15020318/s1, Table S1. Selection of stationary and non-stationary variables based on DIFF values. Figure S1. An overview of RF model computation. Figure S2. Building MGWR model based on GVT test. Figure S3. Regression coefficients interpolated at 1 km resolution. Figure S4. Comparison between the spatial pattern of annual precipitation estimates obtained from the original TRRM (~25 km) and downscaled TRMM (1 km) for different years from 2000–2019. Figure S5. Comparison between the spatial pattern of monthly precipitation estimates obtained after disaggregation of the annual TRRM (~25 km) and downscaled TRMM (1 km) for different months of 2019.

**Author Contributions:** R.N. and A.A.: Conceptualization, Methodology, Software, Writing—first draft. A.A.: Conceptualization, Software, Supervision, Project administration, Writing—review & editing. A.A., J.L., Q.B.P. and D.T.A.: Funding acquisition. Q.B.P. and M.S.: review & editing. J.L., M.S., S.A., A.B., A.M., A.D., D.T.A., S.N.K. and A.E.: review & technical proofreading. All authors have read and agreed to the published version of the manuscript.

**Funding:** This research was funded by [The Belt and Road Special Foundation of the State Key Laboratory of Hydrology-Water Resources and Hydraulic Engineering] grant number [2021491411]. APC were jointly managed by A.A., Q.B.P., and D.T.A.

**Data Availability Statement:** Data processed in this work are available upon request from corresponding author.

**Acknowledgments:** We thank the Pakistan Meteorological Department (PMD) for providing the datasets for this study. We thank Tomoki Nakaya (Department of Geography, Ritsumeikan University, Kyoto, Japan) for developing and providing open source GWR software.

**Conflicts of Interest:** The authors declare no conflict of interest.

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
