# Peer review of "Combining APHRODITE Rain Gauges-Based Precipitation with Downscaled-TRMM Data to Translate High-Resolution Precipitation Estimates in the Indus Basin"

_remotesensing, doi:10.3390/rs15020318_

Round 1
Reviewer 1 Report
Dear Editor and Authors,
My comments can be found in the attached file.

Reviewer 2 Report
Abstract:
The abstract should be able to stand alone as a whole unit separate from the main manuscript. Therefore, acknowledgment of the standard methods used and how your study is different should be included. Also, clearly indicate the study region.
Line 46-47: Switch the abbreviation and long version of the names to follow the standard formatting used in defining abbreviations. E.g., random forest (RF).
Line 50-51: Replace “performed with higher fitting and accuracy” with “performed better on fit and accuracy”.
Line 52-57: Report changes in RMSE and bias values using a single difference value. There are too many numbers, and it currently requires the reader to do the math in their head to truly see if the difference is notable or not. Always make it as easy as possible for your readers to see the result clearly and concisely. For example, instead of the current statement in lines 53-54, it would read like: “… with a reduction in RMSE and bias of X mm/yr and X%, respectively.”
Line 62-66: I have three comments for the last sentence of the abstract:
(1) Replace “satisfactory” with “improved”. Satisfactory would be a bad choice of wording since it denotes barely improved, it’s like giving your results a C for a grade.
(2) For clarity, replace “proposed approach” with the name/abbreviation of approach your results indicated was best. You mentioned testing two methods, reiterate which one was the best to make it very clear for the reader.
(3) Was your study region size global? If not, this is a big leap that it will work well everywhere. Clearly state your study region, then you can say that your results showed promise for that specific region. You can then comment with the caveat that your results "likely have positive implications for use in other catchments, however, further assessment within other regions with varying climates and topography would be required."
Introduction:
Needs some work on the grammar. More importantly, revising is needed for clarity and some sources are missing citations.
Lines 72-74:
Since your research focus is not on extreme precipitation events, I discourage you from focusing this sentence on extreme precipitation. The main issue with using satellite data for the modeling of extremes is the short length of record. So, mentioning it here opens up a whole new issue that you do not address in this paper. Length of record is very important since extreme events are rare! Those of us who model the probability of extreme events require at least 30 years of data to produce good estimates of extreme value distributions at a single site. That is why we typically use spatial statistical methods for extreme event analysis (precipitation, floods, or droughts) to allow for pooling data across sites in the hope of better estimates of the underlying "true" distribution for the region. Even if you are simply pointing out that precipitation products are often used for hazard planning, this is a caveat that needs to be acknowledged as a limitation of satellite-based data - while pointing out that satellite-based data is most often used for near real-time event monitoring and short-term projections (e.g. monitoring an on-going drought). As I said, extremes are a whole different literature review, so focus on near real-time event monitoring and short-term projections for this part of your introduction.
Lines 85-88: The CMORPH, PERSIANN, PERSIAN-CCS, and IMERG products all need citations for their original sources since they are individual products. Cite papers published by the original group(s) that developed the products so they receive the proper recognition.
Lines 98-100: Remove “products in” so the sentence reads “… resolution of grided satellite precipitation products…”
Lines 102-104: These two sentences should be a single sentence if you keep the leading "Although" since it indicates some comparison or caveat is being introduced. Combine the sentences by removing the ". However" at the start of sentence 2.
Lines 105-108: This sentence should be moved to the start of the next paragraph which is all about statistical downscaling. Remove “While”.
Lines 122-123: Define your use of non-stationary and stationary earlier upon first use. In Hydrology we most commonly use the term "non-stationary" to refer to distribution parameters (e.g., mean) that vary in *time*, and "stationary" refers to distribution parameters that do not change over time. So, this could be a confusing use of these terms for readers that are not as familiar with your methods.
Lines 124-125: Change “Therefore, present study utilized the MGWR model…” with “Therefore, this study uses the MGWR model...
Line 141: Typo? GRA performed better than GRA? Which one is meant to read “GDA”? Also, add “according to the literature” to indicate that this is not a result of your own analysis.
Material and Methods
Lines 158-159: Include a small, simple map that shows political boundaries that divide the Indus Basin and the outline of the basin itself for readers that are not familiar with the area. Could be a subplot of Fig. 1.
Lines 167-168: How does the limited number of stations available for validation in zone-2 impact the downscaling results? This should be acknowledged.
Data Sources and Variables
Lines 175-181: Delete all of this. It is the manuscript template notes that were not removed. Needs an introductory paragraph written by the authors to introduce the data used before detailing the sources of data in a paragraph format.
Lines 182-194: Data sources should be detailed in a paragraph format, not as a list of references. Similar to the format of lines 196-214, the authors should describe what data was used, why this particular data was selected, in what part/stage of your analysis was the data employed, and cite references.
Description of Numerical Models
For all formulas in all sections (this one and those that follow), (1) reduce the amount of text within equations. For example, reduce the length of subscripts to abbreviations or single letters. Use variables with subscripts to represent TRMM monthly versus TRMM daily, etc. And (2) describe variables (after the “where…”) in a paragraph format unless you have more than four or so (e.g., Equation 2 makes since to have a list since there are so many variables, the others should be in paragraph format however).
Line 239: Remove “stationary” and “non-stationary” from the equation if it is only for explanatory purposes and does not represent a variable or function within the formula.
Lines 250-251: Only variable held as stationary each time, correct? Why did you use this method to test? Are there other approaches? What about other versions of this… how would it perform when multiple variables were stationary, or how would it perform when only one was non-stationary at a time?
Line 256: How was this threshold (DIFF > 2) determined?
Line 256-257: How does the spatial and temporal resolution of your variables influence the results?
Results
Lines 338-343: Jargon/terminology usage issues that need to be corrected. Adjust the discussion from using the word “performance” to “fit”. The term “predictive performance” cannot be used to describe the model since these are two separate concepts. The TRMM precip data was used to train the models, therefore there should of course be a strong correlation... this correlation does not mean that the models are necessarily predicting precipitation well since you are comparing the model output with the data used to train the model itself. If you truly want to test predictive performance, you will need to use precipitation data that is not used to build the model to check the model. The standard practice for testing model predictive performace is to use 80% of the data (selected at random) to train the model, then the other 20% is used for comparing with model predictions at the held out sites for validation. Often this is done using cross-validation methods (e.g., k-fold cross-validation) that allows for running the model several times with different breaks in the 80/20 random split.
For this whole 3.1 subsection: As is, the comparison in the paper simply shows how much error/bias is being introduced by the environmental variables and model assumptions (aka model fit). Therefore, what methods were used to avoid over-fitting?
Conclusions
The conclusion needs editing. It should be able to stand alone separate from the main text and summarize everything succinctly. This means including again which study region was used and why the study was important compared to other similar studies, in addition to the brief overview of the methods, data, and main results.
